# Inflammasomes—New Contributors to Blood Diseases

**DOI:** 10.3390/ijms23158129

**Published:** 2022-07-23

**Authors:** Jaromir Tomasik, Grzegorz Władysław Basak

**Affiliations:** Department of Hematology, Transplantation and Internal Medicine, Medical University of Warsaw, 02-097 Warsaw, Poland; s080290@student.wum.edu.pl

**Keywords:** inflammasome, hematology, NLR-family-pyrin-domain-containing-3 (NLRP3), NLRP1, NLR-family-CARD-domain-containing-protein-4 (NLRC4), lymphoma, leukemia, myeloma

## Abstract

Inflammasomes are intracellular multimeric complexes that cleave the precursors of the IL-1 family of cytokines and various proteins, found predominantly in cells of hematopoietic origin. They consist of pattern-recognition receptors, adaptor domains, and the enzymatic caspase-1 domain. Inflammasomes become activated upon stimulation by various exogenous and endogenous agents, subsequently promoting and enhancing inflammatory responses. To date, their function has been associated with numerous pathologies. Most recently, many studies have focused on inflammasomes’ contribution to hematological diseases. Due to aberrant expression levels, NLRP3, NLRP1, and NLRC4 inflammasomes were indicated as predominantly involved. The NLRP3 inflammasome correlated with the pathogenesis of non-Hodgkin lymphomas, multiple myeloma, acute myeloid leukemia, lymphoid leukemias, myelodysplastic neoplasms, graft-versus-host-disease, and sickle cell anemia. The NLRP1 inflammasome was associated with myeloma and chronic myeloid leukemia, whereas NLRC4 was associated with hemophagocytic lymphohistiocytosis. Moreover, specific gene variants of the inflammasomes were linked to disease susceptibility. Despite the incomplete understanding of these correlations and the lack of definite conclusions regarding the therapeutic utility of inflammasome inhibitors, the available results provide a valuable basis for clinical applications and precede upcoming breakthroughs in the field of innovative treatments. This review summarizes the latest knowledge on inflammasomes in hematological diseases, indicates the potential limitations of the current research approaches, and presents future perspectives.

## 1. Introduction

Inflammasomes are multiprotein complexes predominantly residing in the cytoplasm of innate immune cells. However, they are also found in lymphocytes and epithelia [1]. To date, their function has been associated with the pathogenesis of various diseases, including metabolic disorders, inflammatory diseases, and neurodegenerative disorders, as well as cancer [2,3]. Moreover, an increasing number of studies suggest their involvement in hematological malignancies.

An inflammasome is defined as a cytoplasmatic multiprotein complex consisting of the following components (Figure 1).

−Pattern-recognition receptors (PRRs)—molecules predominantly belonging to the NOD-like receptor (NLR) family, that detect a wide array of activators. Moreover, absent-in-melanoma-2-like (AIM2-like) receptors and pyrin are also reported as PRRs forming the inflammasomes [4], serving as exclusive exogenous DNA and toxin detectors, respectively [1,5]. The PRR component is essential for the inflammasome to initiate its function.−Apoptosis-associated speck-like protein containing a caspase-recruitment domain (ASC)—an adaptor protein composed of a pyrin domain (PYD), enabling association with the PRR component, and a caspase activation and recruitment domain (CARD), which facilitates the binding of pro-caspase-1 to the PRR–ASC complex. The adaptor protein is absent once PRR contains the CARD domain [4].−Pro-caspase-1—an inactive form of caspase-1, a protease that cleaves the precursors of IL-1β and IL-18 cytokines and other proteins, for instance, gasdermin D [4]. It is activated upon inflammasome assembly, which is elicited by detecting a specific activating signal by PRR and subsequent association of the discussed components [4]. As a result, pro-caspase-1 undergoes autocleavage to form active caspase-1 [6].

The nomenclature of inflammasomes is a derivative of PRRs integrated into the complex [4]. Currently, scientists recognize only a few inflammasomes. These include the well-established NLR-family-pyrin-domain-containing-3 (NLRP3), the NLR-family-pyrin-domain-containing-1 (NLRP1), and the NLR-family-CARD-domain-containing-protein-4 (NLRC4) [4]. However, other PRRs are reported to form alternative inflammasome complexes [4]. The NLRP3 inflammasome is the most extensively studied. Other inflammasomes behave similarly, although some differences may apply [1]. The following paragraph explains the inflammasome priming and activation mechanisms referring to NLRP3.

### Inflammasome Priming and Activation Mechanisms

Activation of the NLRP3 inflammasome requires two steps, as depicted in Figure 2. The first one is priming, which enables the transcription of the inflammasome’s components and caspase-1 substrates [4,7]. It activates specific transduction pathways by certain pathogen-associated molecular patterns (PAMPs) or damage-associated molecular patterns (DAMPs, also known as alarmins). IL-1 family cytokines also elicit priming, often in the positive feedback mechanism [7]. Detection of these signals requires receptors. PRRs often serve this function, transducing the signal via the NF-κB transcription factor, thus acting on gene expression [1,7]. The TLR4/MyD88 pathway is commonly involved in the signaling process [8].

The second step is an assembly of the inflammasome’s components (NLRP3, ASC, and pro-caspase-1) followed by homotypic aggregation of newly formed multimers [1]. This process is triggered by the so-called signal two. Primarily, various activating compounds induce the assembly of the whole protein complex with the subsequent activation of caspase-1. However, NLR does not always detect them directly [4]. Instead, it often responds to cellular events such as ion efflux, lysosomal leakage, and reactive oxygen species (ROS) formation [1,4]. Furthermore, other significant activators are purine metabolites (e.g., extracellular ATP), heat shock proteins (HSPs), and the high mobility group box 1 protein (HMGB1) [8,9]. Notably, ROS response requires the interaction of protein kinase NEK7 with NLR [1], showing the complexity of the signaling networks.

Upon activation, caspase-1 cleaves precursor forms of IL-1β, IL-18, as well as other particles, for instance, gasdermin D (GSMD) [1]. IL-1 family cytokines act on various cell types promoting and enhancing inflammatory responses [10]. They are also reported to contribute to cancer progression [10]. When the inflammasome is hyperactivated, cleavage of GSMD initiates the formation of pores in the cell membrane [11]. Subsequently, it leads to a specific form of cell death, namely pyroptosis [11]. As a cell dies, it releases large amounts of DAMPs, activating an even more robust inflammatory response in the positive feedback loop mechanism [12].

Mechanisms related to inflammasome function seem to play an important role in blood diseases, and the examination of their modulation creates new treatment opportunities. Currently, multiple clinical trials are targeting the IL-1 family in hematological malignancies [8,9], examining predominantly the therapeutic values of either anakinra (antagonist of IL-1R) or canakinumab (IL-1β-neutralizing monoclonal antibody) [8,9]. Notably, it could be reasonable to act on the molecular basis of their release, namely the inflammasome. Furthermore, recent findings showed the NLRP3 inflammasome’s contribution to appropriate hematopoiesis, as it influences the migration and homing of hematopoietic stem cells (HSCs) [11], whereas severe acute respiratory syndrome coronavirus 2 (SARS-CoV-2) has been proposed to damage HSCs in an NLRP3-inflammasome dependent manner [13]. The inflammasome’s role as a disease susceptibility and severity biomarker is also worth consideration. As an increasing number of studies focus on the inflammasomes’ contribution to hematological pathologies, we have decided to summarize the state-of-the-art in inflammasome-aimed research in hematology. Therefore, in the following sections, we discuss several groups of hematological diseases with regard to inflammasomes and propose potential future perspectives for further research.

## 2. Myelodysplastic Neoplasms

Myelodysplastic neoplasms (MDS), formerly known as myelodysplastic syndromes, are cancerous diseases characterized by the overproduction of abnormal cells in bone marrow, which suppress the production of appropriate ones [14]. The disease results from the accumulation of physiology-changing mutations in the hematopoietic stem cell (HSC) population [15]. Therefore, MDS occurs mainly in older patients [15]. Many of them develop AML as the disease evolves [15]. As MDS is often diagnosed in the elderly, some patients are not eligible for high-burden therapies, e.g., alloSCT, due to their health status [15]. Hence, there is a need to design drugs that provide viable treatment options. In recent years, several studies have investigated the role of the inflammasome in MDS pathogenesis.

To date, MDS-associated inflammasome-aimed research has focused exclusively on the NLRP3 inflammasome. The investigations were primarily performed in samples obtained from MDS patients and showed meaningful results. MDS patients’ bone marrow mononuclear cells (BMMNCs) displayed markedly upregulated NLRP3 inflammasome genes [16]. Compared with healthy controls, levels of mRNA encoding caspase-1, NLRP3, IL-1β, and IL-18 were significantly elevated, suggesting strong involvement of the NLRP3 pathway in the pathogenesis [16]. Moreover, apoptosis-related caspase-3 transcripts were not upregulated, suggesting that pyroptosis, not apoptosis is more abundant in MDS pathogenesis [16]. Interestingly, the investigators determined that lower-risk MDS (LR-MDS) samples had higher mRNA expression levels of IL-1 family cytokines than higher-risk MDS (HR-MDS) [16]. Furthermore, in MDS peripheral blood mononuclear cells (PBMCs), another research group confirmed elevated levels of caspase-1 transcripts [17]. Once again, LR-MDS showed higher mRNA concentrations than HR-MDS [17]. Interestingly, bone marrow stromal cells from MDS patients displayed a senescence secretion phenotype, which was attributed to activation of the NLRP3 inflammasome by the S100A9 alarmin [18]. Again, in LR-MDS mesenchymal stromal cells, NLRP3, caspase-1, and IL-1β mRNA expression were higher than in HR-MDS or controls [18]. Notably, NLRP3 inflammasome activation in myelodysplastic neoplasms occurs mainly through the TLR4 pathway [8].

As pyroptosis drives the MDS phenotype [19], the molecules initiating this process have become a field of interest. Thus, the alarmins S100A9 and HMGB1, known to contribute to pyroptosis [16], have been investigated. Subsequently, the researchers determined that plasma concentrations of the abovementioned molecules were significantly increased in MDS samples. Notably, inhibition of the pyroptotic pathway, either by neutralizing S100A9 or blocking NLRP3, abrogated MDS-associated hematopoietic stem cell death and promoted effective hematopoiesis [16]. Another alarmin that has been associated with MDS pathogenesis is oxidized mitochondrial DNA (ox-mtDNA) released upon cell lysis [12]. However, the mechanisms of ox-mtDNA inflammasome induction are yet to be determined [12]. Additionally, a study regarding the SNPs of NLPR3-inflammasome-related genes did not show meaningful implications as very few polymorphisms had possible correlations with MDS pathogenesis [20].

## 3. Lymphomas

Lymphomas are a heterogeneous group of cancers originating from the lymphoid system [21], varying considerably in histopathology, genetics, applied therapies, and prognosis [21]. They are most often characterized by lymphadenopathy, although they can infiltrate the tissues of each organ [22]. Lymphomas are classified into two main groups: B-cell lymphomas and T-cell lymphomas [21]. They are also traditionally divided into non-Hodgkin lymphomas (NHLs) and Hodgkin lymphomas [22]. To date, inflammasome-aimed research has focused exclusively on non-Hodgkin lymphomas. Nevertheless, certain malignancies are still associated with poor therapy outcomes in this diversified group [22]. Therefore, there is a vital need to find new therapeutic targets, and inflammasome research may set the basis for such a discovery.

In B-cell NHLs, comprising 90% of all NHLs [22], several studies have investigated the role of inflammasomes in pathogenesis. The NRLP3 inflammasome and its products may directly promote tumor growth and drug resistance [23]. The activated NLRP3 pathway contributed to the lymphoma’s growth and inhibited apoptosis in a diffuse large B-cell lymphoma (DLBCL) cell line culture [23]. Furthermore, it reduced dexamethasone’s therapeutic effect on lymphoma expansion in vitro [23]. Additionally, samples from newly diagnosed lymphoma patients showed significantly elevated mRNA levels of the IL-18 cytokine, the effector of the inflammasome pathway, compared with controls [23]. Moreover, examination of SNPs revealed that polymorphisms in the IL-18 gene might increase a patient’s susceptibility to developing DLBCL or follicular lymphoma (FL) [24], whereas specific SNPs in the CARD8 gene contribute to decreased survival [24]. Elevated IL-18 was detected in DLBCL patients and correlated positively with PD-L1 expression, a molecule impeding T cell activity [25]. Remarkably, in vivo administration of the NLRP3 inhibitor, MCC950, repressed lymphoma expansion and reduced the concentrations of IL-1β and IL-18, consequently downregulating PD-L1 [25]. A possible explanation is that IL-18 mediates the enhancement of PD-L1 expression by acting on the IFNγ—JAK/STAT pathway in the tumor microenvironment [25]. Intriguingly, the concomitant blockade of IL-18 and PD-L1 resulted in the abolition of anti-tumorigenic effects [25] suggesting that the interactions require further research.

In Burkitt lymphoma triggered by EBV, the downregulated NLRP3 inflammasome could not prevent a latent infection favoring pathogenesis [26]. The TLR4/MyD88 signaling pathway that is involved in signal transduction to the NLRP3 inflammasome was impeded by the NLRP11 molecule [27,28]. Although NLRP11 belongs to the NLR family as NLRP3 does, it possibly contributes to malignancy development by attenuating the immune response [27]. These interactions are an example of the protective impact of the NLRP3 in tumorigenesis and provide evidence for the twofold functions of specific NLRs.

Inflammasomes’ contribution has also been examined in marginal zone lymphomas, specifically mucosa-associated lymphoid tissue (MALT) lymphomas which are associated with Sjögren syndrome (SjS) [29]. Studies determined NLRP3 inflammasome upregulation in SjS patients [30,31] and increased susceptibility to the development of lymphoma [31]. Essentially, mRNA levels of NLRP3, IL-1β, IL-18, caspase-1, and P2 × 7R, components of the NLRP3 inflammasome, were significantly elevated in SjS patients who subsequently developed lymphoma. Therefore, assessment of the NLRP3-axis expression could act as a compelling biomarker for MALT-NHL propensity [31].

In cutaneous T cell lymphoma (CTCL), the NLRP3 protein contributes to pathogenesis in a fascinating mechanism. Unassembled NLRP3, not as a part of the inflammasome complex, moves to the nucleus of CD4+ cells, subsequently activating the IL-4 secretion pathway [32]. IL-4 promotes Th2 T cell shift and decreases the antitumor defense [32]. This mechanism possibly influences the disease’s progression [32]. Another T cell lymphoma in which the NLRP3 inflammasome pathway appears to be relevant is adult T cell lymphoma (ATL) [33]. The researchers reported its upregulation by HBI-8000 (a histone deacetylase inhibitor drug) and following therapeutic effects [33]. In this instance, virally triggered ATL showed susceptibility to NLRP3 activation, which mediates anti-tumor response [33].

## 4. Multiple Myeloma

Multiple Myeloma (MM) is a complex and heterogeneous malignancy characterized by the expansion of monoclonal cancerous plasma cells colonizing the bone marrow [34]. This malignant disease is predominantly incurable [34]. Therefore, there is an urgent need to develop new drugs to improve therapy results. Among potential candidates, inflammasomes emerge as targets for novel therapies.

Research into myeloma cell lines determined that inflammasome-related IL-18 cytokine was critically required for MM progression [35]. Interestingly, IL-1β did not seem relevant compared with IL-18. Moreover, as the NLRP3 inflammasome may be only partially involved in MM pathogenesis, the authors proposed that the NLRP1 inflammasome plays a crucial role in MM progression [35]. Notably, the pathogenesis of MM was linked to the overproduction of IL-18, which contributed to the activation of myeloid-derived suppressor cells (MDSCs). This immunosuppressive switch enabled MM cells to evade immune control [35]. Importantly, neutralizing IL-18 delayed MM progression in a mice MM model [35]. Another study recognized the excessive production of the β2 microglobulin (B2M), a component of MHC 1 class molecules, as an inflammasome-activating and MM promoting factor [36]. In this case, the pathogenesis was associated with the accumulation of B2M aggregates in myeloma-associated macrophages (MAMs) that contributed to the MAMs’ inflammatory response [36]. Subsequent release of both IL-1β and IL-18 resulted from NLRP3 inflammasome activation as analyses showed significantly increased expression of NLRP3, IL-1β, and IL-18 [36]. Furthermore, inhibiting the NLRP3 inflammasome reduced MM progression in the murine model, whereas neutralizing IL-18 impeded MM expansion [36]. Intriguingly, independent gene expression analysis results showed that in newly diagnosed MM patients, transcript levels of NLRP3 and caspase-1 were significantly lower than in the healthy controls [37]. Moreover, NLRP3 expression correlated negatively with the disease stage [37]. The IL-18 induction observed in MM might be only partially dependent on the NLRP3 pathway [37], suggesting the involvement of other inflammasomes, for instance, NLRP1. Additionally, a study investigating gene polymorphisms identified only the CARD8 domain (part of the inflammasome complex) AT genotype as possibly contributing to MM susceptibility, although without impacting overall survival [38]. Moreover, mutual SNPs and gene expression correlations could be either positive or negative [38]. Lastly, the targeting of the discussed pathways has already been described. There is a report that treatment with IL-1 inhibitors decreased the myeloma proliferation rate [39], which encourages additional investigations.

Collectively, the involvement of inflammasomes in the pathogenesis of multiple myeloma is undeniable. The currently available data suggest that IL-18, not IL-1β, is a key player in inflammasome-related disease manifestations.

## 5. Leukemias

Leukemias are blood cancers originating from either myeloid or lymphoid lineages [14,21]. This universal term encompasses a wide spectrum of malignancies, varying in cell maturity, disease course, and therapeutic responses [14,21].

### 5.1. Acute Myeloid Leukemia

Inflammasomes’ contribution to acute myeloid leukemia (AML) is currently the most extensively studied among all hematological diseases. The NLRP3 inflammasome is highly activated and overexpressed in AML cells [40,41]. Bone marrow mononuclear cells (BMMNCs) obtained from newly diagnosed AML patients showed markedly elevated expression of the NLRP3 pathway; however, the results for specific components varied between the studies. Depending on the information source, upregulation of NLRP3, ASC, caspase-1, IL-18, and IL-1β transcripts were reported [40,41]. In peripheral blood mononuclear cells (PBMCs), ASC and IL-18 mRNA levels were significantly upregulated [41]. In both BMMNCs and PBMCs s, all of the above-mentioned components correlated positively with each other and with the white blood count [41]. Moreover, separately estimated levels of IL-18 were also increased [41]. Nevertheless, the impact of IL-18 and IL-1β on AML pathogenesis is not clear, as some studies indicate either of them being more relevant [40,41]. In vivo, the knockout of NLRP3 attenuated leukemic burden; accordingly, neutralizing IL-1β had a similar effect [40]. Intriguingly, NLRP3 inflammasome activation in bone marrow dendritic cells (DC) had an anti-leukemic effect and depended on IL-1β [42], promoting the Th1 response [42]. Moreover, examining HMGB1 alarmin’s impact on AML progression proved its involvement in the pathogenesis via TLR4-mediated NLRP3 inflammasome upregulation in vivo [43].

### 5.2. Chronic Myeloid Leukemia

In chronic myeloid leukemia (CML), overexpression of the NLRP1 inflammasome significantly increased the IL-1β levels and inhibited apoptosis [44]. Collectively, it promoted CML progression both in CML cell lines and PBMCs obtained from patients [44]. Furthermore, upregulation of the NLRP1 inflammasome contributed to the development of imatinib resistance [44]. The mechanism responsible for this phenomenon depends on the endoplasmic reticulum stress-induced activation of NLRP1 [44]. Moreover, IL-18 mRNA levels were elevated in CML patients, whereas IL-1β and NLRP3 transcripts decreased [45]. Furthermore, specific polymorphisms in the NLRP3 inflammasome genes could be associated with susceptibility to CML [45].

### 5.3. Acute Lymphoblastic Leukemia

Remarkable results were obtained in a study regarding the NLRP3 inflammasome’s contribution to acute lymphoblastic leukemia (ALL) in pediatric patients. The most overexpressed genes in glucocorticoid-resistant ALL were NLRP3 and CASP1 (caspase-1) [46]. Moreover, CASP1 and NLRP3 were upregulated in leukemic cells analyzed at relapse compared to the cells analyzed at the initial diagnosis [46]. The researchers administered a caspase-1 inhibitor to the ALL cells and found that the procedure made these cells sensitive to glucocorticoids, thereby overcoming glucocorticoid resistance [46].

Consequently, they demonstrated that caspase-1 cleaved glucocorticoid receptors in their transactivation domain [46]. As glucocorticoids are commonly prescribed medications, these findings are likely to impact resistance management in the future. Additionally, analysis of inflammasome-related genes revealed that IL-1β SNPs in pediatric patients and CARD8 SNPs in adult patients could be potential markers for ALL development susceptibility [47,48].

### 5.4. Chronic Lymphocytic Leukemia

In chronic lymphocytic leukemia (CLL), NLRP3 inflammasome downregulation was shown to contribute to tumorigenesis [49]. Correspondingly, overexpression of the NLRP3 pathway correlated with inhibition of cell proliferation and induction of leukemic cells’ death [49], indicating NLRP3 to be a negative regulator of tumor progression in CLL [49].

## 6. Hemophagocytic Lymphohistiocytosis

Hemophagocytic lymphohistiocytosis (HLH) is a disease characterized by impairment of the immunological system associated with hyperinflammation. HLH is caused by a positive feedback loop in which excessive release of proinflammatory cytokines triggers pathological systemic inflammation, eventually leading to multi-organ failure [50]. It can be subdivided into primary (familial, associated with genetic defect) and secondary HLH (associated with the impairment of NK and cytotoxic T cells), also referred to as macrophage activation syndrome (MAS) in patients with autoimmune disease [50].

As the immune system is impaired in HLH, large amounts of alarmins are released following T cell exhaustion death, activating macrophages [51]. Subsequently, the macrophages undergo pyroptosis, which contributes to the positive feedback loop observed in HLH [51]. Since pyroptosis is the hallmark of inflammasomes’ activation, it is sensible to assume their involvement in the pathogenesis. To date, the NLRC4 inflammasome has been suggested as the leading promoter of HLH [52], whereas the extensively studied NLRP3 does not seem to be prominently engaged in the pathogenesis of HLH. The NLRC4 protein is crucial for HLH as it regulates IL-18 production [53], stimulating IFNγ release [52]. IFNγ is the supreme primer of macrophage activation and, therefore, a fundamental unfavorable factor in HLH, prompting the inflammatory positive feedback loop [51]. Notably, evaluation of samples obtained from MAS patients showed that gene mutations in NLRC4 correlated with a worse disease course [52]. Moreover, enhanced IL-18 production was attributed to increased MAS risk [52,54]. Intriguingly, IL-18 distinguishes secondary HLH/MAS from familial HLH and other autoinflammatory diseases [55].

Collectively, the presented data allow for the conclusion that the NLRC4/IL-18 pathway is a significant player in HLH pathogenesis. To support this view, administration of IL-18BP, the IL-18-blocking molecule, markedly relieved MAS-like symptoms in a patient with confirmed de novo mutation in the NLRC4 gene with concomitant IL-18 serum level elevation [56]. This promising outcome implies a need for further research and possibly clinical trials. Moreover, as secondary HLH may occur following CAR T therapy or EBV infection [55], developing new therapeutics would be especially valuable.

## 7. Graft-Versus-Host Disease

Graft-versus-host disease (GvHD) is a potentially fatal complication of allogeneic hematopoietic stem cell transplantation (alloHSCT) characterized by the reaction of allogenic T cells against the recipient’s tissues [57]. As inflammation caused by conditioning seems to impact GvHD development [57], inflammasomes drew attention as potential contributors to the pathological response.

In vivo experiments proved associations of the NLRP3 inflammasome with GvHD. Its activation by alarmins may lead to overexpression of costimulatory molecules on antigen-presenting cells (APCs), subsequently promoting more robust activation of alloreactive T cells and an aggravated GvHD phenotype [58]. Knocking out specific inflammasome components in graft-recipient mice resulted in amelioration of the GvHD course [59], whereas administration of the NLRP3 inhibitor, glibenclamide, lowered mortality rates in the murine model. Moreover, in samples obtained from alloHSCT patients, increased levels of active caspase-1 and IL-1β were found in circulating leukocytes and intestinal GvHD-related lesions [59]. Interestingly, the contribution of intestinal microbiota may be relevant to the conditioning-induced inflammasome activation [59] as depletion of microbial flora diminished IL-1β levels in murine intestines and skin [59]. A possible explanation is that the release of bacterial products after conditioning-induced mucosal damage promotes a pro-inflammatory response in GvHD [59]. Furthermore, the inflammasome activation in myeloid-derived suppressor cells (MDSCs) impedes their anti-inflammatory function by reducing arginase production [60]. Notably, ex vivo pretreatment of MDSCs with IL-13 prolonged their immunosuppressive activity in vivo [60]. Additionally, inhibiting the NLRP3 inflammasome maintained the MDSCs’ suppressor function [61]. Dendritic cells (DCs) are another group involved in the pathogenesis of GvHD, as they collect antigens and prime allogeneic T cells. DCs’ migration abilities have been assigned to the inflammasome activation. Thus, its impairment may ameliorate GvHD in vivo, which could be mediated by microRNA-155 [62]. As the NLRP3 inflammasome is associated with multiple risks in GvHD [63], studies have examined its role in hepatic complications in a GvHD murine model. After the conditioning, upregulated expression of both IL-1β and IL-18 was found in hepatic tissues concomitantly with elevated NLRP3 and caspase-1 mRNA [64]. Moreover, the increased release of ATP and HMGB1, inflammasome-activating alarmins, was detected [64]. Consequently, administration of the NLRP3 inflammasome inhibitor, BAY 11-7082, alleviated GvHD symptoms and downregulated the expression of NLRP3 and caspase-1 [64]. Accordingly, NLRP3-knockout mice displayed decreased liver infiltration and less severe injuries [63].

Collectively, NLRP3 inflammasome activation in alloHSCT-recipients is attributed to alarmins released after conditioning and contributes to the development of GvHD [59,60].

## 8. Sickle Cell Anemia

Sickle cell anemia (SCA) is a blood disorder characterized by chronic hemolysis resulting from underlying β-globin gene defect [65]. Chronic inflammation is a condition associated with SCA [65], implying the potential involvement of inflammasomes in its pathogenesis.

The NLRP3-inflammasome was found upregulated in platelets obtained from patients as well as in murine models [66]. In vivo, blockade of the inflammasome by MCC950 suppressed caspase-1 activity and decreased platelet aggregation [66]. Consequently, the aggregates were associated with vascular inflammation and occlusion [66]. Moreover, inhibition of NLRP3 reduced platelet build-up in the murine liver [67]. These effects were also achieved by administering ibrutinib, a BTK inhibitor, suggesting BTK kinase’s involvement in the inflammasome regulation [67]. In vitro, coculture of PBMCs with either lysed or intact SCA erythrocytes resulted in the elevated expression of the NLRP3-inflammasome components, IL-1β, and IL-18 [68]. In SCA, NLRP3-inflammasome activation is most likely triggered by the leakage of alarmins from lysed erythrocytes [69]. HGMB1 and heme, acting on TLR4 receptor, seem to play key roles in the upregulation of platelets’ NLRP3-inflammasome activity [70]. Altogether, inflammatory symptoms characteristic of SCA could originate from continuous exposure to hemolysis-related alarmins.

## 9. Inflammasomes and Immunotherapies

The inflammasomes’ pro-tumorigenic role is associated with promoting cell proliferation and inhibition of apoptosis, as well as the immunosuppressive effect on the immune cells [9]. Although inflammasome-targeting immunotherapies have not yet been introduced into clinical practice in hematology, the broad spectrum of interactions provides new opportunities for disease management. The available interventions encompass small-molecule inhibitors, acting directly on the inflammasome pathways, and interleukin blocking agents, acting on the final products of the inflammasomes’ activity. Figure 3 summarizes the inflammasome-modulating approaches discussed in this section.

The impact of the inflammasome signaling is particularly significant concerning CAR T therapy, in which it contributes to cytokine release syndrome (CRS), neurotoxicity, and repression of the T cells [71]. In vitro study on B-lymphoblasts proved that DNA released from lysed cells activated the AIM2 inflammasome in macrophages [71]. Subsequently, IL-1β triggered phenotype switch in the macrophages leading to enhancement of PD-L1 and indoleamine-pyrrole 2,3-dioxygenase (IDO) expression [71]. Consequently, CAR T-cells proliferation was remarkably inhibited [71]. Accordingly, IL-1β released during macrophage response is believed to contribute to CRS onset in vivo [71]. These findings may advocate using anti-PD-L1 antibodies and IDO inhibitors as adjuvants in CAR T therapy, whereas IL-1β blockers could mitigate CRS. Currently, several clinical trials are examining the role of anakinra (antagonist of IL-1R) in mitigating CAR T adverse events [9].

Immune evasion is also characteristic of the inflammasome-mediated pathogenesis of DLBCL [25]. The NLRP3 inflammasome has been shown to regulate immunosuppressive response by increasing PD-L1 expression in the lymphoma cells [25]. Administration of MCC950, small-molecule NLRP3 inhibitor, repressed lymphoma expansion and downregulated PD-L1 [25]. Moreover, MCC950 prevented inflammasome-induced platelet aggregation in sickle cell anemia [66], proving its universal character. Many other inflammasome inhibitors have been tested in the preclinical studies [72], with glibenclamide and BAY 11-7082 shown to mitigate GvHD [59,64]. Altogether, selective small-molecule inhibitors may potentially emerge as valuable hematological drugs.

Another method of inhibiting the inflammasome pathway is the administration of interleukin blocking agents. Apart from anakinra, which is used in managing CAR T adverse events, canakinumab, the IL-1β neutralizing antibody, is being evaluated for the therapeutic effects in MDS and CML [8,9]. Furthermore, administration of IL-18BP, an IL-18-blocking molecule, markedly relieved MAS-like symptoms in an NLRC4-MAS patient [56]. Collectively, interleukin blocking agents appear as suitable treatment options in hematology.

There are several other approaches to repress the inflammasome signaling, yet they require further investigations. Among the most remarkable are gasdermin-targeting drugs which impede pyroptosis and selective caspase-1 inhibitors [73]. Moreover, proteasome inhibitor bortezomib has been shown to selectively inhibit activation of the NLRP1 inflammasome without modulating the NLRP3 inflammasome [73], creating the opportunity to block the NLRP1 signaling in specific diseases.

Besides the targeted drug application, dietary modulation of the inflammasome activity has been reported [74]. Some dietary constituents (e.g., saturated fatty acids), can stimulate the NLRP3 inflammasome activation leading to intestinal inflammation [74]. This finding may set the basis for alleviating intestinal GvHD by changing nutritional habits, yet further investigations are needed.

## 10. Discussion

In recent years, the contribution of inflammasomes to the pathogenesis of various hematological diseases has received researchers’ attention. These studies have provided evidence supporting the involvement of inflammasomes in this area. The best-known inflammasomes, namely NLRP3, NLRP1, and NLRC4 [4], have been shown to influence the course of the diseases discussed in the previous sections (Figure 4). The NLRP3 inflammasome appears especially relevant and is most frequently associated with the pathologies. Its upregulation, measured as the expression levels of its components, has been confirmed in samples obtained from patients suffering from non-Hodgkin lymphomas, acute myeloid leukemia, acute lymphoblastic leukemia, myelodysplastic neoplasms, GvHD, and sickle cell anemia [16,23,30,31,40,41,46,59,66]. Intriguingly, in chronic lymphocytic leukemia samples and Burkitt lymphoma cell lines, downregulation of NLRP3 was associated with cancer progression [26,49]. In either in vitro or in vivo experiments, administration of NLRP3 inhibitors caused amelioration of the disease burden in AML, DLBCL, GvHD, multiple myeloma, and sickle cell anemia [25,36,40,61,67]. Moreover, specific gene variants of the NLRP3 inflammasome were correlated with a susceptibility to developing lymphomas, ALL, CML, and possibly MM [24,38,45,47,48]. Altogether, these results show the potentially universal role of the NLRP3 in the pathogenesis of hematological diseases. Nevertheless, the impact may be twofold as it may act as a pro- or anti-tumorigenic agent, depending on the disease. The role of other inflammasomes has been determined in a few diseases. The NLRP1 inflammasome was linked to multiple myeloma and chronic myeloid leukemia pathogenesis [35,44], whereas the NLRC4 was the key player in secondary HLH [52]. Notably, elevated interleukin-18 levels were confirmed in MM and secondary HLH [35,52,55]. Additionally, gene polymorphisms of the NLRC4 inflammasome correlated with the sHLH course [52]. The findings presented in this manuscript are summarized in detail in Table 1.

Despite many promising results, we are concerned that the conclusions regarding inflammasomes’ contributions could be biased due to potentially incorrect assumptions or limitations in the methodology. Therefore, we indicate sources of possible misconceptions and propose possible solutions.

Primarily, pattern-recognition receptors make up a diversified group, with 23 types of NLRs alone described in humans [1]. They all can putatively be involved in the formation of undiscovered inflammasome types. Nevertheless, the murine genome contains over 30 NLR genes. Thus, there is a potential for the existence of species-unique inflammasomes or inflammasome-related networks, interfering with research results. In this case, data obtained from mice studies may not comply with human characteristics. Furthermore, the gene expression assays performed in the discussed studies usually relied on RT-PCR techniques. Therefore, only the expression of the selected genes was determined, whereas transcript assessment of the putative inflammasomes was infeasible. Additionally, there is a question of whether activation of the inflammasome in vitro adequately resembles the corresponding process in vivo. In cell cultures, artificial inflammasome stimulation is limited to a specific agent, whereas in a living organism, there are various triggers and modifiers of the inflammasome response. Moreover, the impact of specific inflammasome inhibitors on the disease course requires detailed evaluation, as MCC950, a commonly used NLRP3 inhibitor, showed hepatotoxicity [19]. Accordingly, the preclinical use of this compound should be revised as its future application in the clinic is uncertain.

Collectively, the presented concerns highlight the lack of complete understanding of the inflammasomes in hematology. To address this matter, at least partly, we propose the following approach. In each specific hematological disease, determining the actual inflammasome pathways involved in the pathogenesis is crucial for planning accurate interventions and making reliable conclusions. A thorough investigation of the inflammasomes expression in patient-derived samples or cell lines using high throughput techniques could embrace all genes associated with the inflammasome networks. RNA-seq, a novel method of gene expression examination, enables researchers to determine mRNA levels of the whole transcriptome in a single analysis [76]. It could prove whether a specific inflammasome is active and detect PRRs that are potentially upregulated in correlation with the caspase-1 and IL-1 family cytokines. Altogether, this would contribute to understanding the inflammasomes’ role in specific pathologies. Furthermore, the proposed approach becomes more available as the costs of novel research methods decrease. Similar methods could also be applied to mice, increasing the reliability of murine models. Characterization of inflammasome-associated response in hematological diseases could be further improved by implementing single-cell studies (e.g., single-cell RNA-sequencing) to receive a complete picture of gene expression in specific cell subtypes [77]. It would provide insights into how the inflammasome expression in malignant cells correlates with the response of other cells. The information collected could be potentially used as a general predicting factor of drug impact on off-target cells. In addition, the evaluation of extracellular vesicles (EVs) profile would be highly demanded as EVs mediate intercellular communication and are released by inflammasome-activated macrophages [78]. Moreover, repeated gene expression assessments in patients undergoing currently available therapies would undoubtedly provide valuable information on whether a specific inflammasome pathway is affected during the treatment. Analysis of the results obtained from both responding and non-responding patients compared with their initial status could determine the plausibility of targeting inflammasomes in advanced stages of the disease. Additionally, we suggest introducing comprehensive studies, both in vitro and in vivo, investigating various types of inflammasome inhibitors in selected models of hematological diseases. A wide spectrum of the inhibitors tested in a single disease setting could help choose the least toxic molecules for further research and potential clinical application.

Finally, discoveries regarding inflammasomes may become beneficial as new therapeutic agents come into clinical use. For instance, a blockade of inflammasome pathways may attenuate cytokine release syndrome (CRS). CRS is a common complication after CAR T infusion and constitutes serious adverse events associated with novel therapies [79]. Upregulation of inflammasome-associated pathways could function as a predicting factor of CRS onset. Therefore, we advocate for measurements of inflammasome activity in PBMCs obtained from patients anticipating CAR T therapy. It would potentially help prepare for conventional intervention and decrease therapy costs.

## 11. Conclusions

Essentially, inflammasomes have been shown to affect the pathogenesis of various hematological diseases. The promising results of both in vivo and in vitro investigations prove that targeting the inflammasome pathways appears to be a possibly valuable disease-mitigating treatment. Once the inflammasome-related molecular mechanisms are entirely elucidated, and the existing research limitations are managed, the inflammasome-based therapies will likely be applied in clinics with potentially remarkable outcomes.

## Figures and Tables

**Figure 1 ijms-23-08129-f001:**
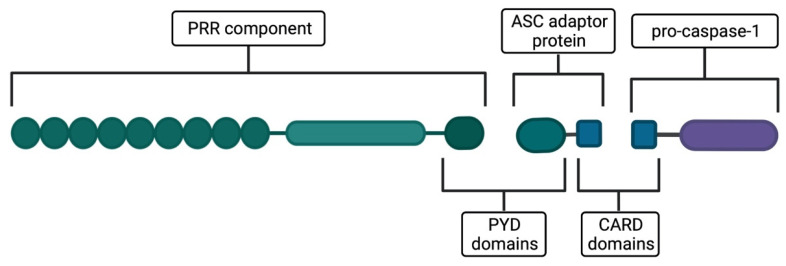
The components of the inflammasome complex—A general scheme. PRR—A component that detects inflammasome activation signals. ASC—An adaptor protein enabling association of PRR with pro-caspase-1. Pro-caspase-1—An effector component of the inflammasome complex. PYD domains—Domains enabling the association of PRR with ASC. CARD domains—Domains enabling the association of pro-caspase-1 with the PRR–ASC complex. Abbreviations: PRR—Pattern-recognition receptor; ASC—Apoptosis-associated speck-like protein containing a caspase-recruitment domain; PYD—Pyrin domain; CARD—Caspase activation and recruitment domain. Created with BioRender.com (accessed on 10 May 2022).

**Figure 2 ijms-23-08129-f002:**
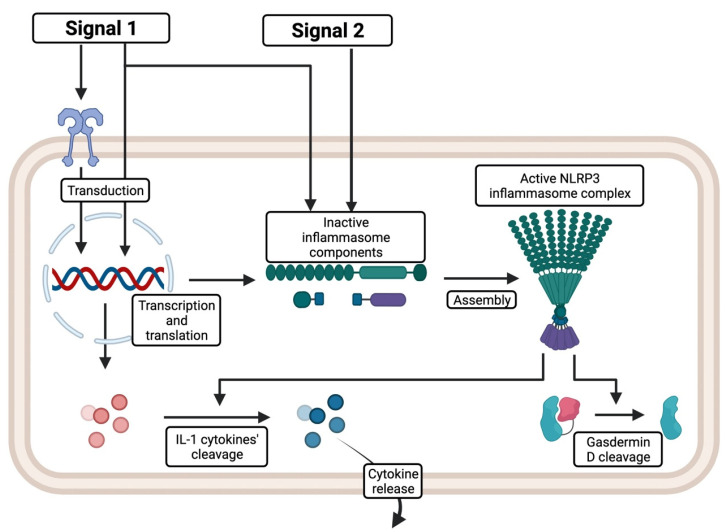
Simplified priming and activation mechanisms of the NLRP3 inflammasome. Signal 1—Priming: detection of specific DAMPs or PAMPs enables transcription and translation of inactive inflammasome components and immature IL-1 family cytokines. Signal 2—Activation: the NLR domain recognizes specific activators that elicit the association of inflammasome components and subsequent assembly of the whole inflammasome complex. Then, the inflammasome cleaves its substrates, predominantly IL-1β and IL-18 cytokines, leading to the inflammatory response. Cleavage of gasdermin D leads to the creation of membrane pores and a specific type of cell death, pyroptosis. Created with BioRender.com (accessed on 11 July 2022).

**Figure 3 ijms-23-08129-f003:**
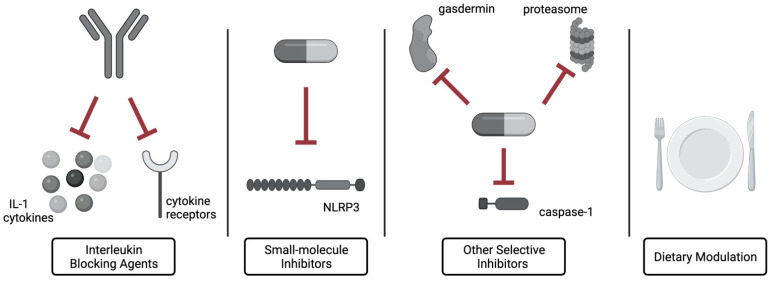
The summary of the inflammasome-modulating approaches. Interleukin blocking agents: antibodies neutralizing cytokines of IL-1 family and their receptors. Small-molecule inhibitors: molecules that directly inhibit the inflammasome components (e.g., MCC950). Other selective inhibitors: molecules selectively inhibiting gasdermin, caspase-1 or proteasome. Dietary modulation: modulating intestinal inflammasome response by change of diet composition. Created with BioRender.com (accessed on 10 July 2022).

**Figure 4 ijms-23-08129-f004:**
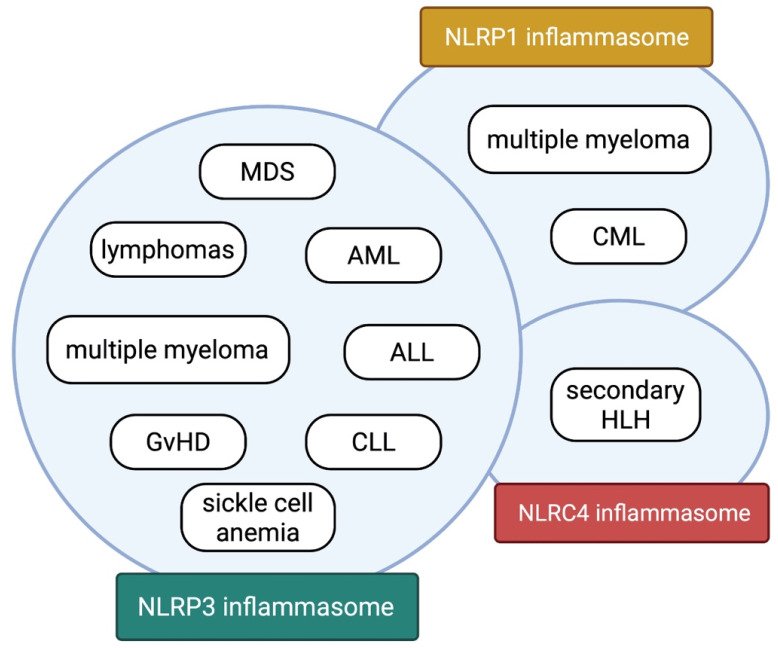
The inflammasomes and their involvement in hematological diseases. The NLRP3 inflammasome contributes to the pathogenesis of numerous hematological diseases: several lymphomas, multiple myeloma, acute myeloid leukemia (AML), acute lymphoblastic leukemia (ALL), chronic lymphocytic leukemia (CLL), myelodysplastic neoplasms (MDS), graft versus host disease (GvHD) and sickle cell anemia. The NLRP1 inflammasome contributes to the pathogenesis of multiple myeloma and chronic myeloid leukemia (CML). The NLRC4 inflammasome contributes to the pathogenesis of secondary hemophagocytic lymphohistiocytosis (secondary HLH). Created with BioRender.com (accessed on 10 May 2022).

**Table 1 ijms-23-08129-t001:** The summary of the state-of-the-art in inflammasome-aimed research in hematological diseases.

Disease	Inflammasome Type	Expression Status	Key Cytokines	Role in Pathogenesis	Related Signaling Pathways	Ref.
Myelodysplastic Neoplasms (MDS)	NLRP3	Upregulated	IL-1β and IL-18	pyroptosis of MDS cells triggers an inflammatory response and promotes the proliferation of neoplastic cells	TLR4/MyD88/IRAK1, IRAK4/TRAF6/NF-κB WNT/β -catenin	[8,12,16,17,18,19]
Diffuse Large B-Cell Lymphoma (DLBCL)	NLRP3	Upregulated	IL-18	NLRP3 inflammasome promotes cell proliferation and apoptosis inhibition and reduces the therapeutic effect of dexamethasone	NLRP3/IL-18/IFN-Y/JAK-STAT/IRF/PD-L1	[23,25]
EBV-positive Burkitt Lymphoma	NLRP3	Downregulated	IL-1β	downregulated NLRP3 inflammasome cannot prevent latent EBV infection NLRP11 molecule represses NLRP3 expression by inhibiting the NF-κB pathway	TLR4/MyD88/TRAF6/NF-κB pathway	[26,27,28]
Marginal Zone Lymphoma	NLRP3	Upregulated	IL-1β and IL-18	lymphomagenesis, NLRP3 inflammasome promotes formation of GC-like structures	P2 × 7R-NLRP3 axis	[30,31]
Adult T-cell Lymphoma (ATL)	NLRP3	Upregulated artificially by HBI-8000 drug	IL-1β	NLRP3 inflammasome prevents latent HTLV-1 infection	not mentioned in the reference	[33]
Multiple Myeloma (MM)	NLRP1, NLRP3	Upregulated	IL-18	NLRP1-dependent activation of MDSCs enables MM cells to evade immune control β2 microglobulin accumulation in myeloma-associated macrophages causes NLRP3-dependent inflammatory response promoting myeloma cell proliferation	unknown	[35,36]
Acute Myeloid Leukemia (AML)	NLRP3	Upregulated	IL-1β and IL-18	NLRP3 inflammasome overexpression in AML cells promotes proliferation and survival NLRP3 inflammasome activation in bone marrow dendritic cells induces Th1 response promoting apoptosis and inhibiting proliferation of AML cells	HMGB1/TLR4, TLR2, RAGE/MyD88/NF-κB IFN-γ/STAT1 pathway in Th1 lymphocytes	[40,42,43]
Chronic Myeloid Leukemia (CML)	NLRP1	Upregulated	IL-1β	NLRP1 inflammasome suppresses apoptosis and promotes proliferation of CML cells and imatinib resistance	IRE1α/CREB/NLRP1 pathway associated with Endoplasmic Reticulum Stress	[44]
Acute Lymphoblastic Leukemia (ALL)	NLRP3	Upregulated	-	caspase-1 cleaves glucocorticoid receptors in their transactivation domain, contributing to glucocorticoid resistance	hypomethylation of NLRP3 and CASP1 (caspase-1 gene) promoters	[46]
Chronic Lymphocytic Leukemia (CLL)	NLRP3	Downregulated	IL-1β	NLRP3 inflammasome downregulation promotes proliferation of CLL cells	P2 × 7R/NLRP3 axis	[49]
Hemophagocytic Lymphohistiocytosis (HLH)	NLRC4	Upregulated	IL-18	T cell exhaustion death causes a release of alarmins activating macrophages, which subsequently undergo pyroptosis contributing to inflammation in the positive feedback loop	unspecified TLR and IFN-γ signaling pathways	[51,52,54,75]
Graft-versus-Host Disease (GvHD)	NLRP3	Upregulated	IL-1β and IL-18	NLRP3 inflammasome activation induces overexpression of costimulatory molecules on APCs, promoting alloreactive T cells response NLRP3 inflammasome activation in MDSCs impedes their anti-inflammatory function by reducing arginase production conditioning-induced mucosal damage promotes the inflammasome-dependent response	TLR4/MyD88/TRIF/NF-κB	[58,59,60,61]
Sickle Cell Anemia (SCA)	NLRP3	Upregulated	IL-1β and IL-18	hemolysis-related alarmins activate NLRP3 inflammasomes in platelets leading to aggregation of platelets	HMGB1/TLR4/MyD88/IRAK4/NF-κB BTK kinase signaling	[66,67,68,69,70]

APC—antigen presenting cell; BTK—Bruton’s tyrosine kinase; CREB—cAMP response element-binding protein; EBV—Epstein–Barr virus; GC—germinal center; HMGB1—high mobility group box 1; HTLV-1—human T-cell lymphotropic virus type 1; IFN-γ—Interferon gamma; IL—interleukin; IRAK—interleukin-1 receptor-associated kinase; IRE1α—inositol-requiring enzyme 1 α; IRF—interferon regulatory factor; JAK—Janus kinase; MDSC—myeloid-derived suppressor cell; MyD88—myeloid differentiation primary response 88; NF-κB—nuclear factor kappa-light-chain-enhancer of activated B cells; PD-L1—programmed death-ligand 1; P2 × 7R—P2X purinoceptor 7; RAGE—receptor for advanced glycation end products; STAT1—signal transducer and activator of transcription 1; TLR—Toll-like receptor; TRAF—TNF receptor associated factor; TRIF—TIR-domain-containing adapter-inducing interferon-β; WNT—wingless-related integration site.

## Data Availability

Not applicable.

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
