# Peer review of "Inflammasomes—New Contributors to Blood Diseases"

_ijms, 2022, doi:10.3390/ijms23158129_

Round 1

Reviewer 1 Report

The review by Jaromir Tomasik and Grzegorz Władysław Basak on the contribution of inflammasomes in blood disease is of interest in a growing topic with contradictory information, is well written and easy to read.

Nevertheless, I would encourage the authors to address the following comments.

 1. Readers will appreciate a summary table of the mayor findings and player for each disease

2. This reviewer do not understand why some diseases have and introductory paragraph and others doesn’t.

3. Is slightly comment here and there the impact of inflammasomes in immune regulation specially associated with immure suppression as well as the impact on immunotherapy like CAR-T cells. It would be of interest if the authors could have a section summarizing this topic and diagram.

4. In the discussion it could also address the use of single cell studies to address some of the issues in understanding this topic.

5. In line 80 it is written “HMBG1”, and it should indicate “HMGB1”

6. In line 461 is indicated “PBMNCs”, I would suggest “PBMCs”.

Reviewer 2 Report

Congratulations on this impressive review. I suggest some minor changes in the attached file

Reviewer 3 Report

Understanding how the inflammation modulate the immune microenvironment at the patients’ bone marrow to promote or inhibit haematological tumours is a clinically relevant topic. Unfortunately, the present manuscript performs a rather general and superficial compilation of findings that relate inflammasome with multiple haematological tumours. This review manuscript needs to be heavily re-structured and sharpened.

Please find some of the comments bellow to improve the review:

Structurally the manuscripts could be improved by first describing the role of inflammasomes in a physiological setting, its importance to mount the inflammatory response (e.g. during injury or infection). Then, a comprehensive explanation on how the alteration of this pathway is specifically exploited by cancer cells in the bone marrow microenvironment is needed. This needs to be accomplished with a clear explanation on how specific inflammasome pathways impact on tumour progression (with a link to tumour hallmarks: increase of tumour proliferation, evading growth suppressors, resisting cell death, inducing angiogenesis, etc rather than a general description that it impacts tumour progression).

 MDS should be discussed prior haematological malignancies (eg. MM, AML, etc. to provide a temporal picture on how inflammasome response changes the environment to promote tumour formation and progression.  

 Several figures need to be added to explain these mechanistic processes in full for each of the haematological diseases.

The authors should have included also a section on the role of the inflammossomes in immunotherapies for these types to cancers (particularly in the case of CAR-T cells therapies for AML and so on).   

 The inflammasome activated cells have distinct responses that influence the microenvironment and thus define the outcome of tumour progression. The authors could also discuss the influence of signalling networks on this process (e.g. secretome of these cells, EV-mediated communication, etc.) is used to perpetuate tumour progression and evasion to therapies.

Round 2

Reviewer 3 Report

The authors did improve some of the aspects pointed by the reviewer improving the overall quality of the review. The manuscript has now satisfactory quality for publication.